# Maternal Salivary miR-423-5p Is Linked to Neonatal Outcomes and Periodontal Status in Cardiovascular-High-Risk Pregnancies

**DOI:** 10.3390/ijms25169087

**Published:** 2024-08-22

**Authors:** Lucia La Sala, Valentina Carlini, Chiara Mandò, Gaia Maria Anelli, Antonio E. Pontiroli, Emilio Trabucchi, Irene Cetin, Silvio Abati

**Affiliations:** 1Department Biomedical Sciences for Health, University of Milan, 20133 Milan, Italy; 2IRCCS MultiMedica, 20138 Milan, Italy; valentina.carlini@multimedica.it; 3Department of Biomedical and Clinical Sciences, University of Milan, 20157 Milan, Italy; chiara.mando@unimi.it (C.M.); gaia.anelli@unimi.it (G.M.A.); 4Department of Health Science, University of Milan, 20146 Milan, Italy; antonio.pontiroli@unimi.it; 5Independent Researcher, 20138 Milan, Italy; trabucchi.emilio@gmail.com; 6Department of Mother, Child and Neonate, IRCCS Cà Granda Ospedale Maggiore Policlinico, 20122 Milan, Italy; 7Department of Dentistry, Vita-Salute San Raffaele University, Milan 20132, Italy; abati.silvio@hsr.it

**Keywords:** miR-423-5p, microRNAs, gingivitis, periodontitis, neonatal outcomes, oral health, pregnancy, cardiovascular risk, obesity, gestational diabetes

## Abstract

Periodontal disease (PD) during pregnancy may trigger systemic inflammation, increasing the risk of developing cardiometabolic disease (CMD). As a consequence, PD may result in the activation of cellular and molecular pathways, affecting the disease course and pregnancy outcome. Although microRNAs (miRNAs) are considered ideal biomarkers for many diseases, few studies have investigated salivary miRNAs and their role in pregnancy or neonatal outcomes. In this study, we sought to investigate the associations between salivary miRNAs of pregnant women with oral diseases and their effects on neonatal outcomes. Eleven (n = 11) salivary miRNAs from a cohort of pregnant women with oral diseases (n = 32; oral health, H; gingivitis, G; and periodontitis, P) were detected using a previous profiling analysis with an FDR < 0.20 and a fold change (FC) < 0.5 or FC > 2 for the most highly expressed miRNAs. Spearman correlations were performed for 11 salivary microRNAs associated with oral-derived inflammation, which could affect neonatal outcomes during pregnancies at risk for cardiometabolic disease (CMD), defined by the presence of a high pregestational BMI. In addition, ROC curves demonstrated the diagnostic accuracy of the markers used. Upregulation of miR-423-5p expression and a decrease in miR-27b-3p expression were detected in the P-group (*p* < 0.05), and ROC analysis revealed the diagnostic accuracy of miR-423-5p for discriminating oral diseases, such as gingivitis versus periodontitis (P vs. G, AUC = 0.78, *p* < 0.05), and for discriminating it from the healthy oral cavity (P vs. H, AUC = 0.9, *p* < 0.01). In addition, miR-27b-3p and miR-622 were also able to discriminate the healthy group from the P-group (AUC = 0.8, *p* < 0.05; AUC = 0.8, *p* < 0.05). miR-483-5p was able to discriminate between the G-group (AUC = 0.9, *p* < 0.01) and the P-group (AUC = 0.8, *p* < 0.05). These data support the role of salivary miRNAs as early biomarkers for neonatal outcomes in pregnant women with periodontal disease at high risk for CMD and suggest that there is cross-talk between salivary miRNAs and subclinical systemic inflammation.

## 1. Introduction

Pregnancy involves physiological changes in which hormones are a trigger for the development of many physiological alterations in metabolism, the immune system, the composition of microbiota, and significant physiological changes in the cardiovascular system [1].

Poor oral hygiene and the loss of healthy dietary habits may prevent the maintenance of well-being throughout pregnancy [2]. Similarly, gingivitis and periodontitis during pregnancy represent a potential risk factor and could trigger adverse neonatal outcomes. A recent meta-analysis evaluated the relationships between maternal periodontal disease and preterm birth (PTB), low birth weight (LBW), and preterm low birth weight [3,4]. The prevalence of periodontal pathology was greater in some groups of pregnant women, reaching a greater prevalence in specific ethnic groups [5,6,7].

Periodontitis is a complex infectious disease preceded by gingivitis, the primary reversible inflammation of gum soft tissues, while periodontitis is characterized by destructive inflammation of gingival tissues with irreversible damage to the alveolar bone, periodontal ligament, and dental cementum [8]. Although periodontitis is a local disease of the oral tissues, recent longitudinal studies have shown that it is associated with the risk of developing cardiovascular disease (CVD) [9] because of its ability to produce inflammatory mediators [10,11]. Systemic inflammation and immune response, with increases in white blood cell count, C-reactive protein, fibrinogen, cell adhesion molecules, and proinflammatory cytokines, may be the core mechanism to explain the association between periodontitis and an increased risk of CVD [12].

In addition, known cardiovascular risk factors, such as hyperglycemia and obesity, could increase the risk of severe outcomes for both pregnant women and newborns. In particular, among obesity-related recurring comorbidities, periodontitis represents an additional important risk factor for pregnancy outcomes [13]. During pregnancy, both obesity and diabetes can cause oral dysbiosis [14], which is the primary cause of periodontal diseases [15], thus aggravating inflammation and the immune response in endothelial vascular dysfunction [16,17].

Along with these mechanisms, growing attention is attributed to the activation of angiogenic mechanisms triggered by inflammation during periodontitis. Angiogenesis is closely related to some biological processes (i.e., embryonic development, reproduction, tissue repair, and wound repair) and may play a key role in the pathogenesis of some inflammatory diseases, such as periodontitis. Anomalous vascularization is one of the most common characteristics in periodontitis leading to progressive inflammation, which results from disrupted periodontal tissues [18,19]. Many biological factors are involved in angiogenic regulation, and most of them are encapsulated in small extracellular vesicles (EVs) with paracrine functions (REF). 

Recently, circulating microRNAs (miRNAs), a class of noncoding RNAs able to regulate gene expression through the inhibition of messenger RNA (mRNA) targets [20], have been proposed as biomarkers for many diseases. Because microRNAs can be detected in biological fluids [21] due to their stability in blood [22], saliva [23], urine, and cerebrospinal fluid [24]—possibly due to their association with RNA-binding proteins (Argonaute2 [Ago2]) or lipoprotein complexes (high-density lipoprotein [HDL]) [25]—or shuttling by membrane-bound extracellular vesicles (EVs), they can mediate cell-to-cell communication between different tissues.

Extracellular vesicles (<200 nm in diameter) from oral biofluids (saliva and gingival crevicular fluid (GCF)) are lipid-encapsulated bilayered vesicles that have recently emerged as a potential source of biomarkers for periodontal disease (gingivitis and periodontitis) due to the cargo of miRNAs, protein, genetic material, and lipids derived from their parent cells. 

miRNAs extracted from EV are more stable and abundant than the not-embedded circulating miRNAs in saliva. Numerous studies of miRNAs have been conducted to identify possible biomarkers that can be used in the diagnosis of periodontal disease [26], in particular because the diagnosis of periodontitis is currently based on clinical rather than etiologic aspects, which can result in limited therapeutic guidance [27]. Hence, it is of great importance to explore potential molecular targets and further conduct effective prevention for patients with periodontitis. Recently, we observed that the salivary extraction of EV-miRNAs was capable of revealing a specific phenotype in obese pregnant women [23]. Furthermore, the advantage of using a saliva-EV-miRNA-based approach is the non-invasiveness of the procedure, coupled with the ability to reflect local (oral) and systemic health due to the peculiarity of the salivary gland [28,29] in digestion and host immune defense and the neuroendocrine regulation of inflammation by salivary polypeptides. The surrounding microvessels can favor the exchange of circulating molecules that can influence saliva composition, reflecting the systemic status and the risk to develop cardiovascular-related diseases (CVDs). 

In this study, we sought to explore the utility of salivary miRNAs for periodontal disease in pregnant women. In addition, we describe the relationship between neonatal outcomes and oral health during pregnancy and investigate the role of several salivary miRNAs, which are potential diagnostic tools for periodontitis and may be able to predict potential adverse pregnancy outcomes.

## 2. Results

### 2.1. Demographic Data

Characteristics of the enrolled pregnant population divided into three groups (healthy—H, gingivitis—G, and periodontitis—P) are reported in Table 1. The mean age was significantly higher in P compared to H pregnant women (*p* < 0.05). In the G-group, the 2 h plasma glucose (2 h-PG, mg/dL) and the plaque index (%) were significantly greater than those in the H-group (*p* < 0.05). In the P-group, parameters, such as the number of teeth (*p* < 0.01), BOP (*p* < 0.0001), PPD (*p* < 0.0001), the presence of calculus (%; *p* < 0.0001), and DMFT (*p* < 0.05), were significantly greater than those in the H-group. The number of teeth (*p* < 0.05), BOP (*p* < 0.05), PPD (*p* < 0.0001), the presence of calculus (%; *p* < 0.001), and DMFT (*p* < 0.05) were significantly greater in the P-group than in the G-group.

With regard to placental data, placental thickness (cm) was significantly increased in the P-group, with a *p* value of 0.05.

### 2.2. Pregestational Body Mass Index (pre-BMI), Oral Status, and Neonatal Outcomes

The correlation analysis between pregestational body mass index (pre-BMI) and maternal oral outcomes, such as the number of teeth (negative, r = −0.5, *p* < 0.01), BOP (positive, *p* < 0.05), and % plaque (positive, *p* < 0.05), suggested that obesity has a strong impact on maternal oral health. With respect to neonatal outcomes, we found that pre-BMI was positively correlated with placental weight (r = 0.5, *p* < 0.01) and placental thickness (r = 0.4, *p* < 0.05) and negatively correlated with the feto/placental (F/P) weight ratio (r = −0.4, *p* < 0.05), as shown in Figure 1.

As obesity is considered a strong contributor to low tissue saturation (hypoxic tissues [30]), we sought to correlate pre-BMI with umbilical oxygenation data at delivery, and we detected negative correlations (pO_2_UA mmHg, r = −0.5, *p* < 0.01; satO_2_ content UA%, r = −0.6, *p* < 0.001; O_2_ content UA%, r = −0.6, *p* < 0.001), suggesting that obesity also has a negative impact on neonatal outcomes in line with previous results [31]. No other maternal characteristics correlated with the placental and neonatal parameters.

As a consequence of the inflammation that occurs in relation to pre-BMI [32], the number of teeth (nT) is an indirect indicator of periodontitis (P-disease) and is correlated with a greater CHD risk [33]. In this study, nT was negatively correlated with placental weight (r = −0.6, *p* < 0.01) and positively correlated with the F/P weight ratio (r = 0.6, *p* < 0.05). Moreover, BOP as an indicator of gingivitis and/or periodontitis is related to local inflammation, and its correlation with placental thickness (r = 0.4, *p* < 0.05) is in line with the findings of other studies reporting a thickened placenta as a result of several inflammatory conditions [34,35,36] and systemic vascular disorders [37,38].

### 2.3. Selection Strategy for Relevant miRNAs

To determine the influence of P-disease on the level of miRNAs in pregnant women, we performed profiling [23]. Using a linear regression model adjusted for pregestational-BMI, age, gestational age, and presence of dysglycemia in three groups of pregnant with or without oral disease, and applying the False Discovery Rate (FDR) < 0.2 estimation with *p*-value < 0.05, we selected the eleven most significantly expressed (n = 11) salivary miRNAs: miR-127, miR-23a-5p, miR-27b-5p, miR-27b-3p, miR-124a, miR-423-5p, miR-483-5p, miR-532, miR-551b, miR-622, and miR7-1-1 (Appendix A). 

All miRNAs were normalized using the means of two classical normalizers, U6-RNU48, via the comparative Ct method in pregnant women. Then, alterations in these parameters were analyzed among the following groups: oral healthy pregnant (H), pregnant with gingivitis (G), and pregnant with periodontitis (P) (Figure 1).

Enrichment KEGG analysis and GO enrichment analysis were performed using miRPath v.3 through the Diana web tool (http://diana.imis.athena-innovation.gr) (Appendix A).

### 2.4. Differentially Expressed miRNAs in the P-Group: miR-423-5p and miR-27b-3p

In the P-group, we observed an alteration in the expression of two microRNAs. miR-423-5p was increased (*p* < 0.05), whereas miR-27b-3p was decreased (*p* < 0.05), both with respect to those in the H-group (Figure 1), suggesting that these miRs play key roles in P-disease. Moreover, miR-423-5p significantly discriminated the P-group from the G-group (P vs. G, AUC = 0.78, *p* < 0.05; Figure 2).

We found a significant correlation between miR-423-5p and several maternal periodontal health parameters (PPD, *p* < 0.01; and % calculus, *p* = 0.05) and neonatal outcomes (neonatal weight, *p* < 0.05; placental thickness, *p* < 0.05; placental weight, *p* = 0.05; and LGA, *p* = 0.06), as shown in Table 2A–C. Additionally, we found a strong negative correlation between miR-423-5p and oxygen saturation (satO_2_ content UV, *p* = 0.03, r = −0.9), as shown in Table 2. We also observed an increase in miR-423-5p in the P-group and reduced saturation in the newborn group (Table 2), suggesting that miR-423-5p plays a role in tissue hypoxia.

However, miR-27b-3p seems to be more strongly linked to oral health. It is correlated with BOP (*p* < 0.001), % calculus (*p* < 0.01), and the bleeding index (*p* < 0.001).

### 2.5. Correlations between miRNAs and Maternal Periodontal Health Parameters

In this study, we found (Table 2C) that miR-127 and 483-5p were correlated with BOP (*p* < 0.05 and *p* < 0.001, respectively). miR-23a-5p was correlated with nT (*p* < 0.05) and PPD (*p* < 0.01).

Interestingly, miR-483-5p expression levels are reduced in both gingivitis (*p* = 0.003) and periodontitis (*p* < 0.01) (Figure 1). The significant correlations between miR-483-5p and BOP (*p* < 0.01) are indicative of a strict correlation with the alteration of gingival vascularization.

miR-27b-3p expression levels are significantly reduced in periodontitis (*p* < 0.05) and related to periodontal disease parameters, such as BOP (*p* < 0.001) and % calculus (*p* < 0.01). 

### 2.6. Diagnostic Specificity and Sensitivity of Salivary miR-423-5p, miR-483-5p, and miR-27b-3p for Periodontal Diseases during Pregnancy

To study the diagnostic accuracy of salivary miR-423-5p, miR-483-5p, and miR-27b-3p as surrogate biomarkers for the gingivitis and periodontitis status, a receiving operator characteristic (ROC) curve was drawn. The data demonstrated the diagnostic accuracy of miR-423-5p (P vs. G, AUC = 0.8; *p* = 0.04; H vs. P, AUC = 0.9; *p* = 0.007) as a biomarker of the P-group (Figure 2). Importantly, the AUC of miR-423-5p exhibited higher values in discriminating P from H. We are able to discriminate between miR-423-5p in pregnant patients with periodontitis and those with a high CV risk. We also assessed the diagnostic performance, specificity, and sensitivity of miR-483-5p and miR-27b-3p in discriminating the P and G phenotypes (Figure 2). 

## 3. Materials and Methods

### 3.1. Study Participants

Thirty-two (n = 32) pregnant women were enrolled in the antenatal clinic at the Obstetric Unit of the L. Sacco Hospital (ASST Fatebenefratelli-Sacco) in Milan.

The study protocol was approved by the hospital ethical committee (Prot. N. 469/2010/52/AP). Written informed consent was obtained to collect personal data and biological samples. Pregnant women were enrolled during the first trimester with regular clinical follow-up during all trimesters. The oral and periodontal examination and characterization of this population have been previously reported [17]. Only Caucasian women who had singleton spontaneous pregnancies and aged between 18 and 40 years were enrolled. The exclusion criteria were maternal and fetal infections, fetal malformations, chromosomal disorders, maternal alcohol/drug abuse, and pregestational body mass index (BMI) < 18.5. Oral and periodontal health evaluations were performed in the third trimester, as previously reported [17,39]. Pregnant patients were classified as follows: 1) healthy with overall good oral care (no dental plaque or calculus, no gingival bleeding on probing (BOP), no pocket depth over 3 mm on probing; and (2) periodontally diseased patients with poor oral care, with (i) gingivitis (dental plaque and calculus in ≥6 teeth and gingival bleeding ≤ 5 teeth) and/or (ii) periodontitis (probing pocket depth ≥ 4 mm, dental plaque, calculus and bleeding ≥ 6 teeth). The two gingival conditions were grouped together. 

Neonatal data, placental biometry, and biochemical measurements were collected at the time of delivery. Placental weight was obtained after discarding membranes and the umbilical cord.

Umbilical venous and arterial blood samples were obtained from a doubly clamped segment of the cord immediately after fetal delivery, and blood gases were measured on a GEM Premier 3000 (Instrumentation Laboratory, Brussels, Belgium). 

### 3.2. Biological Sample Collection

*Saliva collection:* Saliva sampling was obtained during the third trimester without any stimulation (passive drooling technique). Enrolled women (n = 32) were asked to refrain from eating, drinking sugary or alcoholic beverages, smoking, and performing invasive oral care procedures for at least 1 h before collection. They were then asked to rinse their mouth (1 min) with a physiological solution to remove any food residues, not to swallow for 3 min, and then spit in a sterile tube. The collected samples were stored at −80 °C.

### 3.3. Isolation of Extracellular Vesicles (EVs) and miRNA Detection

Saliva samples (1 mL) were centrifuged at 4000× *g* for 30 min at 4 °C to remove any cell debris and aggregates. The supernatants were ultracentrifuged at 110,000× *g* for 75 min at 4 °C to pellet the EVs and then stored at −20 °C.

EV-miRNA isolation was performed with a combination of the miRNeasy Kit and RNeasy Cleanup Kit (Qiagen, Hilden, Germany) according to the manufacturer’s protocol. They were eluted in 20 μL of nuclease-free water and stored at −80 °C until use. EV-miRNA quality and integrity were assessed through the “2100 Bioanalyzer RNA system” (Appendix A) with the Pico Kit (Agilent Technologies, Santa Clara, CA, USA), and the concentration (ng/μL) was assessed using a Quantus Fluorometer (Promega, Milan, Italy). Fifteen nanograms of reverse-transcribed (RT) miRNA was preamplified (16 cycles) and analyzed using q–PCR with the QuantStudio™ 12K Flex OpenArray^®^ Platform (Applied Biosystems, Waltham, MA, USA). miRNA expression was determined using the relative 2^−∆∆Ct^ method.

### 3.4. Data Analysis and Statistics

Statistical analysis was performed using GraphPAD Prism v10. The data distributions were checked with histograms and the Kolmogorov–Smirnov test. When a variable is normally distributed, the data are presented as the mean ± SD. Qualitative variables are expressed as numbers and percentages. Comparisons were performed using either the Wilcoxon test (signed-rank) or the Friedman test (nonparametric tests for paired data). Comparisons between two groups were performed by using either Student’s *t* test or the Mann–Whitney *U* test (two-tailed) for normally or nonnormally distributed variables, respectively. Proportions were compared by using the χ^2^ and Fisher’s exact tests. The relationships between variables were analyzed using Pearson’s or Spearman’s correlation coefficient for parametric or nonparametric data, respectively. Receiver operating characteristic (ROC) curves were used to establish promising biomarkers for P. The area under the ROC curve (AUC) measures the performance of the biomarkers. Excellent biomarkers had an AUC of 1.0; good biomarkers had an AUC > 0.80. Using these criteria, we list in Figure 3 the set of possible good salivary biomarkers for periodontal disease. Statistical significance was set at the 95% level (*p* < 0.05).

## 4. Discussion

The increasing evidence of the ability of miRNAs to provide relevant biological information about cells and tissues supports the fascinating hypothesis that miRNAs act as endocrine and paracrine messengers. In this scenario, salivary miRNAs [23] can be considered a promising low-cost new source of molecular biomarkers for a plethora of human diseases. In particular, for an oral health assessment tool, which often is based on clinical diagnosis measuring periodontal tissue loss, the discovery of new classes of diagnostic biomarkers, such as the saliva-EV-miRNA-based approach, is of critical importance for detecting periodontitis in real time.

The role of miRNAs in periodontitis remains to be elucidated. Many studies have compared inflamed gingival tissues with healthy tissues using different methods of miRNA detection and profiling, most of which are based on miRNA microarrays, resulting in a large variability in miRNA expression patterns [40]. In this study, we investigated correlations among the major miRNAs analyzed in a cohort of pregnant women with oral diseases who were at high risk of developing cardiometabolic diseases (CMDs), defined by the evaluation of their pregestational BMI (Table 1) and the presence of dysglycemia. Recent studies have revealed that in utero exposure to maternal obesity and gestational diabetes mellitus (GDM) are associated with an increased risk of obesity and CMD in future adults [41]. At the same time, obesogenic epigenetic changes play a major role in fetal reprogramming during development [42]. Most maternal risk factors are associated with the adaptive growth response of the placenta [38]. It seems that placental weight (PW) and placental morphologic characteristics, such as placental thickness (PT), may be pivotal in the effects of maternal overweight and weight gain during pregnancy on neonatal weight and fat mass [43]. We observed a direct correlation among PW and PT and a negative correlation between the F/P weight ratio and pregestational BMI (Figure 3) that might be co-attributed to the presence of subclinical inflammation associated with oral pathologies.

The role of miRNAs in the host immune response and inflammation through the regulation of inflammatory cytokines has been proposed to be associated with oral pathologies. Interestingly, based on miRNA profiling of overweight patients with periodontitis, obesity is thought to be a risk factor for periodontal tissue inflammation [44]. In this study, the correlations highlighted the possible influence of maternal salivary miRNAs on neonatal outcomes. The biological mechanisms for this assumption could be explained by the fact that miRNAs influence the fetus through actions on placental growth and function, but they may also communicate directly with the fetal cells and tissues, thus influencing neonatal outcomes, such as birth weight and fetal growth [45]. Our study supports this possibility, as the increased expression of salivary miR-423-5p in pregnant women with periodontitis was correlated with worse neonatal outcomes in terms of neonatal weight (kg), placental weight, and placental thickness. Growing evidence indicates that the mechanism behind the morphological placental features/changes could be driven by the activation of angiogenic patterns, such as oxygen tension and specific growth factors, delineating the role of hypoxia and inflammation as triggers for angiogenesis in periodontitis [46]. Salivary miR-423-5p could be a molecular biomarker for assessing angiogenic activation in periodontitis. Functional in vitro studies revealed the mechanisms behind the activation of miR-423-5p transcription during angiogenesis: miR-423-5p is enhanced by E2F1 (E2F Transcription Factor 1) and modulated by hypoxic factors, such as HIF1a, and growth factors, such as VEGF (vascular endothelial growth factor) [47]. Our observations of the increased placental thickness and increases in miR-423-5p expression levels in the periodontitis group support the hypothesis of an angiogenic milieu in these patients. In addition, miR-423-5p seems to be an important marker for maternal and perinatal outcomes. It was found in maternal plasma as an early predictor of preeclampsia (PE) at an early gestational age because it was demonstrated that it is involved in the regulation of trophoblast cell migration and invasion, and the dysregulation of these processes may promote the development of PE [48] through the activation of Wnt signaling, PI3K/AKT signaling, and focal adhesion molecule signaling activated by integrin clustering and providing the reorganization of actin cytoskeleton [48].

In addition, we detected a positive correlation between miR-423-5p and PPD, suggesting that miR-423-5p is strongly linked to periodontal tissue destruction, which is the outcome of periodontal disease. The correlations found are suggestive of the specific expression of these miRNAs in saliva samples (Figure 3).

Recently, miR-423-5p and miR-23a-3p have been found to be upregulated in crevicular fluids of periodontopathic patients, suggesting their effective involvement in the oral diseases mediated by inflammation [49]. Referring to the different expression levels between miR-423-5p (up) and miR-27b-3p (down) in the periodontitis group, it is noticeable that both miRNAs could activate the same molecular pathway, as shown from the relationship found in the miRNA cluster dendogram (Appendix A, KEGG analysis; miRPath v.3; www.dianatools). To our knowledge, we have reported for the first time the reduced miR-27b-3p expression levels in salivary samples of periodontitis pregnancies. The literature data evidenced that the miR27a/b cluster may be involved in oral inflammation, with an involvement of miR-27a-3p downregulation in patients with pulpal inflammation [50]. As reported in Figure 3, miR-27b-3p correlated with placental thickness, also suggesting its involvement in placental development. Furthermore, Peng et al. [51], performing gain- and loss-of-function experiments, found miR-27b-3p to be reduced in the process of osteogenic differentiation and impeded this process by inhibiting Sp7 expression. These data might contribute to current understandings of the biological link between miRNAs, periodontitis [52], and cardiometabolic risk. In particular, it has been reported that miR-27b-3p plays a crucial role in the regulation and maintenance of metabolic homeostasis [53]. Our results suggest that miR-23a, -27b-3p, and -5p are components of the same miRNA family with similar evolutions but different expression patterns, whose function is likely linked to angiogenic genes. Due to their plasticity, they could modulate differentially the entire angiogenetic process by targeting multiple antiangiogenic genes. miR-27b is highly expressed in endothelial cells (ECs) and enhances angiogenesis by promoting EC proliferation and migration and targeting Sprouty2 (Spry2) and semaphoring 6A and 6D (Sema6A-6D) in response to VEGF [54], among others. 

Our study has some limitations due to the intrinsic difficulties. First, this work has a relatively small sample size, limiting the possibility of generalizing the obtained considerations and of using these data to define the diagnostic sensitivity and specificity of the studied miRNAs. Notwithstanding these limitations, the analyzed population was well-defined by inclusion and exclusion criteria, clinical characteristics that reduced any additional clinical bias. Secondly, an evaluation of functional studies would be helpful to understand the role of miRNAs in the cross-talk between oral health and pregnancy outcomes. In addition, an analysis of the proteome, transcriptome, and microbiome would allow for obtaining a complete picture of their role in periodontal disease.

## 5. Conclusions

In this work, we highlighted the potential ability of salivary miRNAs as novel molecular biomarkers. Many studies have attempted to explain their possible biological role in many processes, from oral diseases to placental development, cell differentiation, proliferation, apoptosis, invasion/migration, and angiogenesis.

Altogether, our data strongly suggest that miR-423-5p could be an early predictor of abnormalities linked to pregnancy and periodontal disease. We were able to discriminate between miR-423-5p in pregnant patients with a high CV risk and periodontitis and gingivitis. Measuring salivary miR-423-5p may be a useful new biomarker for periodontitis in pregnant patients. 

Overall, these findings suggest that an epigenetic approach is a feasible strategy for identifying new biomarkers that juxtapose canonical measures in the early detection of cardiometabolic abnormalities in saliva samples. 

## Figures and Tables

**Figure 1 ijms-25-09087-f001:**
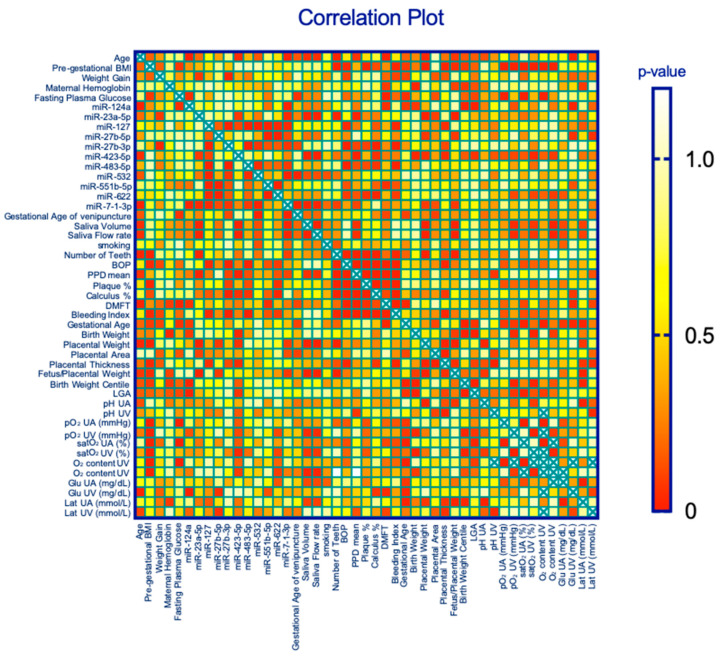
Correlation plot of the subject parameters: miRNAs and oral/neonatal outcomes.

**Figure 2 ijms-25-09087-f002:**
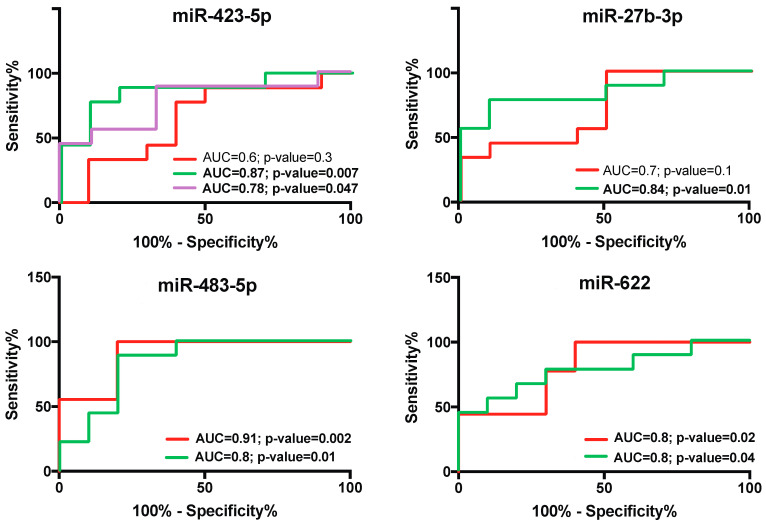
Receiver operating characteristic (ROC) curves and area under the ROC curve (AUC) generated for sensitivity analysis showing the diagnostic performance of salivary miRNAs. ROC curves of salivary miRNAs in detecting oral health status in pregnant women. The area under the curve (AUC) is reported as the performance measure. The red line represents the ROC curve for the gingivitis status, the green line represents the ROC curve for periodontitis, and the violet line represents the ROC curve of the discriminatory ability to distinguish periodontitis from gingivitis.

**Figure 3 ijms-25-09087-f003:**
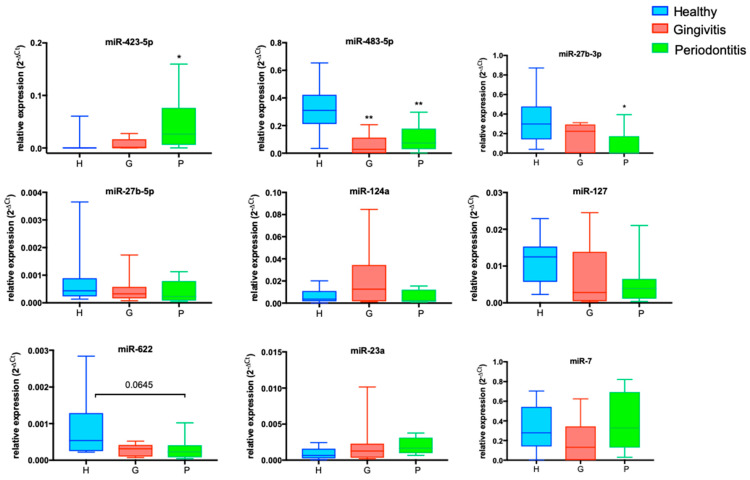
miRNA expression levels in gingivitis and periodontitis phenotypes. Box and whiskers of logistic regression models adjusted for diabetes, age, and smoking status for the probability of having a gingivitis (G) or periodontitis (P) phenotype. All miRNAs were normalized using the means of two classical normalizers, U6-RNU48, used for the comparative Ct method in pregnant women (n = 30). The data are presented as the means (±SDs). Kruskal–Wallis test followed by post hoc tests; * *p* < 0.05, ** *p* < 0.01.The expression levels of nine of the eleven miRNAs are shown in the graphs. miR-423-5p was upregulated in the P-group compared with the H-group (*p* < 0.05); miR-483-5p and miR-27b-3p were downregulated in the P-group compared with the H-group (*p* < 0.05 and *p* < 0.01, respectively); miR-124a tended to increase in the G-group; miR-127 tended to decrease in the P-group; miR-622 tended to decrease in both the G- and P-groups; and miR-23a tended to increase in the P-group.

**Table 1 ijms-25-09087-t001:** Characteristics of the enrolled subjects (pregnant women).

	Healthy (H) (n = 10)	With Gingivitis (G) (n = 9)	With Periodontitis (P) (n = 9)	Overall *p* Value	H-G *p* Value	H-P *p* Value	G-P *p* Value
** *Maternal data* **							
Age ^B^, years	29.6 ± 4.93	31.9 ± 3.65	34.8 ± 3.87	0.04	0.5	0.033	0.3
Pregestational BMI ^BC^, kg/m^2^	26.5 ± 7.59	28.3 ± 7.85	31.4 ± 11.52	0.51	0.9	0.5	0.7
NW, n (%) OB, n (%)	6 (60.0) 4 (40.0)	4 (44.4) 5 (55.6)	4 (44.4) 5 (55.6)				
Pregestational hypertension ^C^, n (%)	2 (20.0)	0 (0.0)	2 (22.2)				
Gestational diabetes (GDM) ^C^, n (%)	1 (10.0)	5 (55.6)	3 (33.3)				
Fasting glycemia ^B^, mg/dl	87.7 ± 12.1	84.9 ± 9.90	89.5 ± 15.7	0.74	0.9	0.9	0.7
1-hour postload glycemia ^B^, mg/dl	111.8 ± 37.5	136.0 ± 39.1	132.3 ± 52.6	0.43	0.5	0.6	0.9
2-hour postload glycemia ^B^, mg/dl	92.0 ± 21.3	138.7 ± 46.3	108.0 ± 43.6	0.04	0.03	0.6	0.2
Gestational age at sampling ^A^, weeks	33.1 ± 2.33	33.3 ± 2.62	33.3 ± 1.95	0.97	0.99	0.99	1.0
Number of teeth ^B^	27.9 ± 0.33	27.0 ± 1.50	24.3 ± 3.54	0.004	0.6	0.004	0.04
Gingival BOP ^B^, % sites	6.35 ± 14.5	23.2 ± 19.3	59.6 ± 25.7	<0.0001	0.18	<0.0001	0.002
Gingival PPD ^B^, mm	2.01 ± 0.12	2.25 ± 0.25	3.30 ± 0.61	<0.0001	0.37	<0.0001	<0.0001
Plaque index ^A^, %	11.1 ± 18.2	39.7 ± 18.0	68.0 ± 33.3	0.0001	0.04	<0.0001	0.04
Calculus ^A^, %	9.92 ± 16.5	29.3 ± 14.1	69.2 ± 25.2	<0.0001	0.09	<0.0001	0.0005
Dental health index (DMFT) ^A^	5.56 ± 3.68	5.33 ± 3.00	9.78 ± 4.15	0.02	0.99	0.04	0.04
** *Maternal, Placental, and Neonatal Data at Delivery* **							
Gestational age ^A^, weeks	39.4 ± 1.03	39.5 ± 0.91	39.9 ± 1.27	0.6	0.9	0.6	0.7
Maternal GWG ^B^, kg	10.8 ± 7.00	11.8 ± 3.93	11.8 ± 5.02	0.9	0.9	0.9	1
Neonatal weight ^A^, g	3369.5 ± 380.2	3357.8 ± 315.0	3480.6 ± 353.4	0.7	0.99	0.8	0.7
Neonatal weight centile ^AC^	50.7 ± 29.7	55.0 ± 27.0	56.7 ± 31.5	0.9	0.95	0.9	0.99
AGA, n (%) LGA, n (%)	8 (80.0) 2 (20.0)	8 (88.9) 1 (11.1)	6 (66.7) 3 (33.3)				
Neonatal sex ^C^ M, n (%) F, n (%)	6 (60.0) 4 (40.0)	3 (33.3) 6 (66.7)	3 (33.3) 6 (66.7)				
Placental weight ^A^, g	450.8 ± 81.4	470.0 ± 77.8	526.7 ± 104.8	0.2	0.9	0.2	0.4
F/P weight ratio ^B^	7.71 ± 1.62	7.32 ± 1.40	6.81 ± 1.40	0.4	0.8	0.4	0.7
Placental area ^B^, cm^2^	278.9 ± 55.3	276.4 ± 57.4	248.2 ± 39.1	0.4	1.0	0.4	0.5
Placental thickness ^A^, cm	1.67 ± 0.45	1.74 ± 0.32	2.19 ± 0.57	0.04	0.9	0.05	0.1

The data are expressed as the mean ± standard deviation (SD) and were analyzed according to their distribution with independent samples. ^A^ ANOVA, ^B^ Kruskal–Wallis test, or ^C^ Freeman–Halton extension of Fisher’s exact probability test; the statistical significance from post hoc tests is indicated in bold. BMI: body mass index; NW: normal weight; OB: obese; GDM: gestational diabetes mellitus; BOP: bleeding on probing; PPD: probing pocket depth; DMTF: decayed, missing, and filled teeth; GWG: gestational weight gain; M: male; F: female; F/P: feto/placental.

**Table 2 ijms-25-09087-t002:** Correlations between miRNAs and neonatal outcomes (2A) and maternal oral health (2B).

(**A**)								
**Neonatal Outcomes**	**mir-23a-5p**	**mir-423-5p**	**mir-127**	**mir-124a**				
	*p* value	*p* value	*p* value	*p* value				
Neonatal Weight (kg)	-	0.034	-	-				
Placental Weight (kg)	0.003	0.055	-	-				
Placenta Thickness (cm)	-	0.043	0.050	-				
Fetus–Placental Weight (Ratio)	0.013	-	-	-				
Weight Newborn Centile	-	-	-	0.016				
LGA Newborn	-	0.063	-	0.007				
(**B**)								
**Neonatal Outcomes**	**Maternal Oral Outcomes**							
	N teeth	BOP	DMFT					
Gestational Age (GA)	-	-	0.03					
r (rho)	-	-	−0.39					
Placental Weight (PW, kg)	0.00	-	-					
r (rho)	−0.59	-	-					
Placental Thickness (PT, cm)	-	0.03	-					
r (rho)	-	0.44	-					
Weight Fetus/Placenta	0.00	-	-					
r (rho)	0.59	-	-					
(**C**)								
**Maternal Oral Outcomes**	**miRs**							
	23a-5p	423-5p	7-1-3p	127	27b-3p	483-5p	551b-5p	622
	*p* value	*p* value	*p* value	*p* value	*p* value	*p* value	*p* value	*p* value
Saliva Volume	**0.005**	**0.001**	**0.009**	-	-	-	-	-
Flow Rate (mL/min)	**0.005**	**0.001**	**0.009**	-	-	-	-	-
Number of Teeth (nT)	**0.016**	-	-	-	-	-	-	-
BOP	-	-	-	**0.033**	**0.000**	**0.004**	-	-
PPD	**0.008**	**0.001**	-	-	-	-	-	-
% Plaque	-	-	-	-		-	-	0.053
% Calculus	-	0.056	-	-	**0.006**	-	-	**0.019**
DMFT	-	-	-	-	-	-	**0.051**	-

## Data Availability

The datasets generated during the current study are not publicly available but are available from the corresponding author upon reasonable request.

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
