# Peer review of "Maternal Salivary miR-423-5p Is Linked to Neonatal Outcomes and Periodontal Status in Cardiovascular-High-Risk Pregnancies"

_ijms, 2024, doi:10.3390/ijms25169087_

Round 1

Reviewer 1 Report

Comments and Suggestions for Authors

In this manuscript, Authors have investigated the associations between salivary miRNAs of pregnant women with oral diseases and their effects on neonatal outcomes. Although impressive, the authors are requested to address the following issues to make the manuscript technically sound.

Major comments:

1.        Authors are encouraged to clearly emphasize the significance of the miRNA-based approach and its advantages over existing diagnostic tools for measuring oral health and its association with cardiovascular disease (CVD).  Authors are requested to clearly explain how this approach can be more robust compared to current diagnostic methods.

2.        Authors are requested to provide more details on how the particular miRNAs (miR-127, miR-23a-5p, miR-27b-5p, miR-27b-3p, miR-124a, miR-423-5p, miR-483-5p, miR-532, miR-551b, miR-622, miR7-1-1.) were selected. Clearly explain the rationale behind selecting specific miRNAs and their possible role in oral diseases during pregnancy and its association with CVD.  

3.        In the introduction, Authors have mentioned, "Recently, we observed that the salivary extraction of EV-miRNAs was capable of revealing a specific phenotype in obese pregnant women [21]." Authors are requested to provide some more details describing a clear co-relation between specific miRNA and their target genes, and how they possibly lead to disease condition.

4.        Authors are requested to include RNA integrity data (derived from Bioanalyzer) along with RIN score to support that miRNA isolated and investigated here have baseline quality

5.        The discussion lacks specific biological effects of these miRNAs could have on the pregnant women and fetus. Although authors have provided a preliminary explanation on miRNA423's association with neonatal health, discussion on how other miRNA selected in this study, especially miRNA27b-3p might have role in disease progression is missing. Authors are requested to provide a clear insight on the possible underlying molecular and biological mechanisms.

Author Response

COMMENT REVIEWER 1

In this manuscript, Authors have investigated the associations between salivary miRNAs of pregnant women with oral diseases and their effects on neonatal outcomes. Although impressive, the authors are requested to address the following issues to make the manuscript technically sound.

Major comments:

  1. Authors are encouraged to clearly emphasize the significance of the miRNA-based approach and its advantages over existing diagnostic tools for measuring oral health and its association with cardiovascular disease (CVD).  Authors are requested to clearly explain how this approach can be more robust compared to current diagnostic methods.

Thank you for pointing this out. In introduction section we have emphasized the significance of salivary miRNA-based approach for measuring oral health and its association with cardiovascular disease.

  1. Authors are requested to provide more details on how the particular miRNAs (miR-127, miR-23a-5p, miR-27b-5p, miR-27b-3p, miR-124a, miR-423-5p, miR-483-5p, miR-532, miR-551b, miR-622, miR7-1-1.) were selected. Clearly explain the rationale behind selecting specific miRNAs and their possible role in oral diseases during pregnancy and its association with CVD.  

Thank you for this important point. We are agreed to provide more details of selection method: at page 5, selection strategy was implemented with the required information. We accordingly provided the selecting methods for our 11 miRNAs and we tried to provide an explanation of their possible involvement in periodontal diseases and pregnancy outcomes, including neonatal outcomes. Furthermore, in Discussion section we added a rational for all miRNAs and their involvement in periodontitis/pregnancies and neonatal outcomes.

In particular, we provided adjunctive literature references to explain the role of such miRNAs in relation to disease studied in this work.

  1. In the introduction, Authors have mentioned, "Recently, we observed that the salivary extraction of EV-miRNAs was capable of revealing a specific phenotype in obese pregnant women [21]." Authors are requested to provide some more details describing a clear co-relation between specific miRNA and their target genes, and how they possibly lead to disease condition.

Thank you for this important point. Firstly, we changed this reference with another due to a software-error. We have also provided an explanation of possible relations and co-relations with miRNAs and their putative target genes, mostly related to angiogenetic phenotype. 

  1. Authors are requested to include RNA integrity data (derived from Bioanalyzer) along with RIN score to support that miRNA isolated and investigated here have baseline quality

Thank you for pointing this out. We uploaded a supplementary file with all informations required and inserted in the manuscript a supplementary figure 1.

  1. The discussion lacks specific biological effects of these miRNAs could have on the pregnant women and fetus. Although authors have provided a preliminary explanation on miRNA423's association with neonatal health, discussion on how other miRNA selected in this study, especially miRNA27b-3p might have role in disease progression is missing. Authors are requested to provide a clear insight on the possible underlying molecular and biological mechanisms.

We fully agree with this point and thank the reviewer for this important question. We have implemented the discussion with the explanation of the interplay between miR-423-5p increase and miR-27b-3p decrease on the biological significance. In addition, we have inserted in Discussion also a possible molecular and biological mechanism.

Reviewer 2 Report

Comments and Suggestions for Authors

Sala et al. conducted a nice work which investigated the association between salivary microRNAs in pregnant women with oral diseases and their effects on neonatal outcomes, particularly in those with high pregestational BMI. This study presents novel evidence for the role of salivary miRNAs, particularly miR-423-5p, as a potential biomarker for periodontal disease. While the findings are promising, further research with larger cohorts is needed to validate these results and explore their clinical applications.

There are a few concerns regarding the manuscript.

1. In the method part, you mentioned  that the salivary miRNAs were derived from extracellular vesicles. Where do you think these salivary extracellular vesicles come from? What is the point of isolating miRNAs from extracellular vesicles instead of directly from saliva?

2.Salivary miR-423-5p quite showed some potential on diagnosing periodontal diseases during pregnancy. What do you think would be the possible mechanism? How could these salivary miRNAs affect the neonatal outcomes ?

3. I didn’t see the comparison between high BMI and normal BMI group, it’s therefore hard to conclude the miRNAs were linked to cardiovascular high-risk pregnancies.

4. In the discussion part, it’s necessary to mention the limitations of the study.

5. Italian words in the manuscript should be replaced by English words. Intensive English edits is necessary.

Comments on the Quality of English Language

Extensive editing of English language required

Author Response

COMMENT REVIEWER 2

Sala et al. conducted a nice work which investigated the association between salivary microRNAs in pregnant women with oral diseases and their effects on neonatal outcomes, particularly in those with high pregestational BMI. This study presents novel evidence for the role of salivary miRNAs, particularly miR-423-5p, as a potential biomarker for periodontal disease. While the findings are promising, further research with larger cohorts is needed to validate these results and explore their clinical applications.

Thank you for the comment. We certainly agree about the need to increase our cohort of patients and develop an experimental study in which validate these microRNAs for clinical practice.

There are a few concerns regarding the manuscript.

  1. In the method part, you mentioned that the salivary miRNAs were derived from extracellular vesicles. Where do you think these salivary extracellular vesiclescome from? What is the point of isolating miRNAs from extracellular vesicles instead of directly from saliva?

Reply to Q1 and Q2: About the atavic point about the origin of the salivary extracellular vesicles, we think they derived from oral cavity in which converges all molecules, proteins derived from oral epithelium and endothelium of microvessels surrounding salivary glands. As explained in the text, these vessel networks allowing for the cross-talk between distant organs vehiculating many molecules, through the circulation, in distant body district, such as placenta.

In fact, in crevicular fluids of periodontitis were found miRNA-423-5p and -23a-5p together with a plethora of pro-inflammatory proteins REF 45. It can be assumed that these miRNAs through the circulation could reach the placenta, a high-vascularized tissue connected to fetal tissues, determining neonatal outcomes.

miRNA extracted from EV are more stable and abundant than the not-embedded circulating miRNAs in saliva. For this statement we extracted miRNAs from saliva-EV.

  1. Salivary miR-423-5p quite showed some potential on diagnosing periodontal diseases during pregnancy. What do you think would be the possible mechanism? How could these salivary miRNAs affect the neonatal outcomes ? How could these salivary miRNAs affect the neonatal outcomes ?

Due to the anatomy of salivary glands and the surrounding microvessels, salivary miRNAs could entry into the circulation and affects neonatal outcomes such as placental weight and thickness.

Several studies have investigated the link between miRNA expression profiles in maternal and fetal tissues and adverse maternal and perinatal outcomes. Most of these studies have focused on preeclampsia reporting dysregulated expression of some miRNA species in relation to these outcomes.

For istance, miR-423-5p seems to be an important marker for maternal and perinatal outcomes. It was found in maternal plasma miR-423-5p as an early predictor of preeclampsia.

  1. I didn’t see the comparison between high BMI and normal BMI group, it’s therefore hard to conclude the miRNAs were linked to cardiovascular high-risk pregnancies.

In table 1, we listed all variables measured comprised pregestational BMI that in periodontitis was found increased with respect to other groups. Obesity is notoriously a high-risk factor for CVD.

  1. In the discussion part, it’s necessary to mention the limitations of the study.

Thank you for this point. We added this part before the conclusions.

  1. Italian words in the manuscript should be replaced by English words. Intensive English edits is necessary.

Round 2

Reviewer 1 Report

Comments and Suggestions for Authors

Authors have addressed all comments. Accept in present form. 

Reviewer 2 Report

Comments and Suggestions for Authors

I have no more comments on the manuscript